# Spinal Cord Impairment in Anti-Mag Neuropathy: Evidence from Somatosensory Evoked Potentials

**DOI:** 10.3390/brainsci10050282

**Published:** 2020-05-08

**Authors:** Marilisa Boscarino, Jacopo Lanzone, Lorenzo Ricci, Mario Tombini, Vincenzo Di Lazzaro, Giovanni Assenza

**Affiliations:** Unit of Neurology, Neurophysiology, Neurobiology, Department of Medicine, University Campus Bio-Medico of Rome, via Álvaro del Portillo, 21, 00128 Rome, Italy; m.boscarino@unicampus.it (M.B.); j.lanzone@unicampus.it (J.L.); lorenzo.ricci@unicampus.it (L.R.); m.tombini@unicampus.it (M.T.); v.dilazzaro@unicampus.it (V.D.L.)

**Keywords:** anti-MAG, spinal cord, neuropathy, somatosensory evoked potentials

## Abstract

Anti-Myelin Associated Glycoprotein (anti-MAG) neurological involvement classically manifests as a peripheral neuropathy with prominent sensitive symptoms. We describe a case report of a patient with positive anti-MAG antibodies presenting with clinical and neurophysiological evidence of spinal cord impairment. A 69-year-old woman came to our attention with subacute onset of dysesthesias at lower limbs and ataxia. Blood routine tests and hematological work-up led to a diagnosis of monoclonal gammopathy of undetermined significance. High titers of anti-MAG antibodies was revealed (34,594.70 BTU/mL, normal range 0–1000). Nerve conduction studies (NCS) ruled out a polyneuropathy at lower limbs. Somatosensory evoked potentials (SSEPs) showed prolonged central conduction time (CTT) at lower limbs, suggesting a dorsal column damage. Brain and spinal cord Magnetic Resonance Imaging (MRI) did not reveal any significant lesion. Analysis of cerebrospinal fluid (CSF) evidenced an albumin-cytologic dissociation. She was treated with corticosteroids with temporary remission of sensory symptoms and normalization of CTT. Subsequently, she developed a multineuropathy which was successfully treated with Rituximab. We discuss the potential role of anti-MAG antibodies in the pathophysiology of dorsal column impairment and the clinical usefulness of SSEPs in monitoring the evolution of anti-MAG neuropathy.

## 1. Introduction

Chronic neuropathies are a common cause of neurological disability worldwide and it is estimated that about one-fifth of these patients do not receive an appropriate etiologic diagnosis thus demonstrating the clinical challenge in dealing with this type of patients [1].

Ten percent of patients with polyneuropathy of unknown cause have a monoclonal gammopathy, mostly represented by IgM paraglobulinemia [2]. In patients with IgM monoclonal gammopathy associated neuropathy, serum anti-myelin associated glycoprotein (anti-MAG) antibodies are frequently detected at high titers [3].

Clinical picture of anti-MAG associated neuropathy includes distal dysesthesias, ataxia and tremor, with mild motor symptoms developing only in later stages [4]. Nerve conduction studies (NCS) are crucial in diagnosis and frequently show a symmetrical demyelinating neuropathy with distal prominence (distal acquired demyelinating symmetric neuropathy, DADS). However, one third of patients may present with sensori-motor distal and proximal demyelinating signs, which fulfill diagnostic criteria for chronic inflammatory demyelinating polyneuropathy (CIDP). A minority of cases may also show asymmetric and multifocal neuropathy impairments [5]. Anti-MAG neuropathy may also start with subclinical neuropathic impairment which cannot be identified by standard NCS. In atypical presentations of anti-MAG neuropathy, somatosensory evoked potentials (SSEPs) may provide some clinical value because they are able to detect minor proximal neuronal damages which are hardly evidenced by NCS. Indeed, although there are several reports of SSEPs studies in anti-MAG neuropathy patients, the main finding reported is a conduction impairment of the cauda, while a spinal dorsal column damage was never testified [6].

In the present manuscript we report an atypical presentation of anti-MAG neuropathy where initial symptoms and negative NCS studies led to perform SSEPs demonstrating a dorsal column impairment before the development of polyneuropathy.

## 2. Case Description

A 69-year-old Caucasian woman presented at our attention with subacute onset (more than eight weeks) of “constricting” dysesthesias at lower limbs and imbalance. Neurological examination demonstrated loss of sense of vibration and proprioception at lower limbs, ataxic gait, positive Romberg sign and tetrahyporeflexia. Muscle strength of the four limbs was normal. Her remote pathological history encompassed a breast cancer diagnosis (IIB-stadium) thirteen years before, treated with surgery, radiotherapy and chemotherapy (cyclophosphamide, pharmorubicin and anastrozole). Clinical and radiological follow-up were unremarkable, except for a persistently weak positivity of the Ca 15.3 marker. Four limbs NCS evidenced a slight bilateral carpal tunnel syndrome with no signs of distal or proximal neural damage at lower limbs Table 1.

Brain and spinal cord MRI showed an enlargement of cervical and lumbar roots without gadolinium enhancement as for chronic roots inflammation, in absence of intramedullary lesions Figure 1.

SSEPs of upper limbs, performed at median nerves, showed normal values Figure 2A; on the contrary, SSEPs of lower limbs evidenced normal latency values of spinal stationary potentials (N22, 21.1 ms right, 20.1 ms left; upper normal limit in our laboratory: 25.8 ms), but a pathological latency of P40 cortical potentials (right 45.2 ms, left 46.5 ms; upper limit in our laboratory: 43.9 ms) with a prolonged central conduction time (CCT, 24.1 ms right, 26.5 ms left; upper normal limit of our laboratory: 21.3 ms), suggesting a dorsal column involvement Figure 2B.

Blood routine tests (including thyroid hormones, glucose test, vitamin B12, folate, lactic acid, inflammatory and infectious serology, onconeural and antigangliosides antibodies, autoimmune screening) and hematological work-up (also including whole body X-ray and bone biopsy) led to a diagnosis of IgM kappa-light chain Monoclonal Gammopathy of Undetermined Significance (MGUS). High serum titer of anti-MAG antibodies was revealed (34,594.70 BTU/mL, normal range 0–1000), with a weak positivity of anti-sulfated glucuronyl paragloboside (SGPG) antibodies (1.20 U/mL, normal range <1).

Analysis of cerebrospinal fluid (CSF) evidenced an albumin-cytologic dissociation, without oligoclonal bands (pattern I, Link index of 0.5). CSF bacterial and fungal cultures and neurotrophic viral nucleic acid searches were negative. The patient was treated with an intravenous high-dose corticosteroid therapy (methylprednisolone 250 mg/die for three days) and reported an improvement of sensory and imbalance symptoms in concomitant with a significant reduction of anti-MAG titer (7520 BTU/mL) Figure 3.

SSEPs were performed again and showed, at lower limbs, bilateral normalization of CCT (right 17.2 ms, left 16.1 ms;) in contrast with pathological N22 latencies (right 28.9 ms, left 29.4 ms) Figure 2B, thus suggesting a regression of the dorsal column damage and the development of peripheral neuropathy.

After few months, she complained a relapse of sensitive and ataxic symptoms and was treated with a cycle of plasmapheresis followed by transitory clinical remission. NCS were repeated and showed a sensorimotor multineuropathy with demyelinating damage of both proximal and distal segments of lower limbs Table 1. The lack of a sustainable peripheral venous access catheterization contraindicated a chronic treatment with plasmapheresis and, consequently, a four-weekly infusion of 375 mg/m^2^ Rituximab was started.

Patient reported a subjective improvement of “constricting” dysesthesias at lower limbs and neurological examination showed a negative Romberg sign and gait stability. Last NCS registered an improvement of demyelinating damage (decrease of tibial nerve and peroneal nerve distance motor latency on both sides and reduction of tibial F-wave latencies) Table 1. Clinical and NCS neuropathy improvement endured till the last follow-up at 20 months.

## 3. Discussion

Spinal cord dorsal column impairment is an atypical presentation of anti-MAG antibodies neuropathy. Our patient presented with ataxic gait and constriction neuropathic pain at lower limbs strongly suggesting a spinal cord involvement with no evidence of peripheral neuropathy at NCS. SSEPs were clinically useful to support our hypothesis of dorsal column damage in absence of any radiological or CSF alteration congruous with this finding. Clinical symptoms and SSEPs spinal abnormalities were ameliorated by corticosteroids therapy in parallel with a significant reduction of anti-MAG antibodies titer. Furthermore, the high titer of anti-MAG antibodies was the only pathological finding of a complete and prolonged clinical work up. Thus, we assume that anti-MAG antibodies could have played a role in the pathophysiology of dorsal column damage in our patient before the onset of peripheral neuropathy.

The transmembrane glycoprotein MAG is a constituent of both central and peripheral nervous system myelin. MAG is concentrated in periaxonal Schwann cell (SC) membranes and paranodal loops of the peripheral nerve myelin where it acts as an adhesion molecule for interactions between SC and axons [7]. In animal models of spinal trauma and dysmyelination, as well as human neurological diseases, SCs move into the Central Nervous System (CNS) and remyelinate CNS axons, resulting in the repair of axonal function and even in the reversal of neurological impairments. In these models, MAG strongly binds SCs, inhibits their migration and induces their death in vivo [8], thus impairing spinal myelin repair.

The role of anti-MAG antibodies in CNS damage is unclear. In animals, Sergott and colleagues [9] injected a monoclonal antibody to MAG with 20% guinea pig serum into mammalian optic nerves and produced in vivo demyelination associated with three ultrastructural patterns of myelin injury (widened lamellae, myelin vesiculation and cell-associated myelin damage), which were observed neuropathologically in multiple sclerosis and in demyelinating peripheral neuropathy associated with plasma cell dyscrasias. These results establish the ability of anti-MAG to produce in vivo demyelination of mammalian CNS and indicate that a single antibody directed against a specific myelin component may initiate multiple types of myelin damage. 

CNS targets of anti-MAG antibodies were found also beyond the myelin. Denton et al. [10] screened human sera from patients with IgM paraproteinemia with frozen sections of the rat cerebellum and observed a characteristic pattern of neuronal staining interesting the perikarya and immediate processes of large neurons of the deep cerebellar nuclei (the dentate, embolliform and globose). Further examination of the staining pattern in the whole rodent brain indicates that the paraprotein is highly selective and binds to specific groups of neurons located in the granular layer of the cerebellar cortex, namely the Lugaro cells. They observed a similar staining pattern also in the cerebral cortex, principally confined to the pyramidal cells of layers 3 and 5, predominantly in the temporo-parietal sensory-motor cortex, but also in lateral reticular thalamic neurons, cells of the nucleus ruber, nucleus lateralis lemniscus ventralis, lateralis vestibularis, cochlearis dorsalis and scattered pyramidal cells of the hippocampus. Interestingly, a similar staining pattern was observed with mouse and rabbit cerebellum. These experimental models provide evidence of the involvement of the MAG protein in repairing processes following spinal damage and the possibility for anti-MAG antibodies to penetrate and target CNS epitopes.

We also performed a literature search in order to find all the evidences of a central nervous system, and in particular, spinal cord involvement in anti-MAG positive patients. In humans, anti-MAG antibodies bind both to central and peripheral nervous system myelin, although with different affinity [11] and their immunostaining is highly variables among patients, thus contributing to explain the heterogeneity of clinical and neurophysiological presentation. We found few and controversial case reports where anti-MAG antibodies seem to play a role in CNS disorders. Sotgiu and colleagues [12] reported a clinical case of a 44-year-old man where an initial diagnosis of multiple sclerosis was made on the basis of symptoms (hands and feet paresthaesias, mild paraparesis), brain and spinal cord MRI showing six T2-weighted and fluid-attenuated inversion recovery (FLAIR)-weighted hyperintense white matter lesions without contrast enhancement, resembling those of multiple sclerosis and CSF showing the presence of ten oligoclonal bands (OCB). In that case, paresthesias on hands and feet started nine years before the slow development of gait ataxia and footdrop. Following neurological examinations, performed consequently, led to a concomitant diagnosis of a prevalently demyelinating sensory motor polyneuropathy and a high title of anti-MAG antibodies was detected, thus configuring a picture of MGUS associated IgM anti-MAG polyneuropathy. The patient did not respond to a high-dose of intravenous methylprednisolone, but after three cycles of Intravenous Immunoglobulins (IVIGs) he experienced clinical benefits accompanied by electrophysiological improvement on NCS, the regression of two T2-weighted hyperintense brain MRI lesions (while spinal MRI was unchanged), the decrease of CSF OCBs number (from 10 to 4) and the normalization of both CSF and serum anti-MAG antibody levels. Benedetti et al. [13] described a 59-year-old man with an 18-year history of multiple sclerosis presenting with an unusually rapid progression of paraparesis with hypopallesthesia and areflexia in four limbs, where following neurophysiological and serologic examinations led to the diagnosis of MGUS associated anti-MAG polyneuropathy. The patient was consequently treated with Rituximab. They observed clinical and neurophysiological improvement of polyneuropathy at the expense of two multiple sclerosis relapses. The authors ascribed deterioration of the patient’s condition to multiple sclerosis relapses because of the prompt response to corticosteroids to which anti-MAG neuropathy is unresponsive. Twelve months after the starting of Rituximab, clinical conditions remained stationary, anti-MAG-specific and total IgMs decreased significantly and, surprisingly, their patient, who was OCBs positive at diagnosis, was found to be persistently OCBs negative after the onset of anti-MAG polyneuropathy and treatment with Rituximab. This finding is unusual because OCBs represent a stable abnormality in MS. They supposed that monoclonal anti-MAG IgM could have primed mechanisms that interfere with persistent OCB production by oligoclonal plasma cells. In patients with multiple myeloma/MGUS, it has been reported the existence of an autoimmune inhibitory network leading to the arrest in B-cell differentiation and consequent humoral immune deficiency, possibly sustaining, in this case, the disappearance of IgG OCBs. Ziset al. [14] described five anti-MAG positive patients with IgM monoclonal gammopathy (among these, only four patients have a diagnosis of neuropathy), with evidence of cerebellar rather than just sensory ataxia. They treated them with Rituximab with benefit and concluded that anti-MAG antibodies may be involved in the pathogenesis of idiopathic sporadic ataxias, irrespective of the evidence of peripheral neuropathy. These case reports provide some suggestions of a possible anti-MAG involvement in CNS pathologies and in particular in demyelinating disorders of CNS, however the pathogenetic role of anti-MAG antibodies in spinal pathologies remains controversial and needs further proofs [15].

In this scenario, SSEPs may have a crucial role in contemporary exploring both the peripheral nerve and the dorsal column functionality. There are several studies investigating the clinical utility of SSEPs in diagnosing and classifying CIDP patients [16]. SSEPs demonstrated their utility in revealing proximal sensory nerve dysfunction particularly in patients with demyelinating sensorimotor polyneuropathy not evidenced by conventional nerve conduction studies.

SSEP studies confirmed proximal sensory nerve dysfunction in pure sensory ataxic groups of patients, leading to a correct diagnosis and treatment. Furthermore, in patients with DADS neuropathy, SSEPs were able to differentiate anti-MAG-positive from anti-MAG-negative patients by revealing predominantly distal slowing in the latter group.

Overall, these findings suggest that SSEP studies may be a crucial ancillary neurophysiological technique to conventional NCS in classifying CIDP and, in particular, anti-MAG neuropathy.

## 4. Conclusions

We report the first case of a symptomatic patient with anti-MAG antibodies presenting with a dorsal column impairment, as documented by SSEP studies, before the onset of peripheral neuropathy. We speculate that this clinical presentation and neurophysiological findings may be expression of a pathogenetic role played by anti-MAG antibodies in demyelination of both peripheral and central nervous system. Current literature in animal models and humans does not seem to deny this thesis, even if further studies, which combine immunological research and clinical observation, are necessary to validate our hypothesis. We encourage the use of SSEPs in classifying and monitoring the evolution of anti-MAG sensory neuropathy because of their ability to simultaneously explore central and peripheral sensitive nerve conduction.

## Figures and Tables

**Figure 1 brainsci-10-00282-f001:**
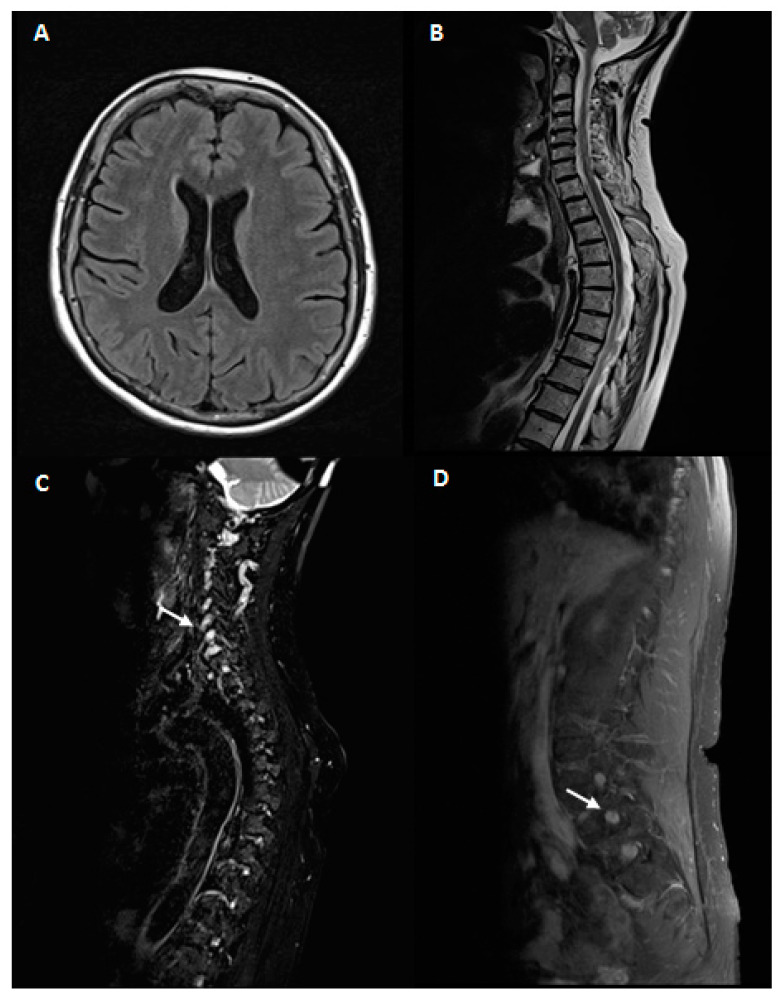
Magnetic resonance imaging(MRI)study. 1.5 Tesla brain and spinal cord MRI. A, B: brain MRI axial T2 fluid-attenuated inversion recovery (FLAIR) (**A**) and spinal cord MRI sagittal T2 fast spin echo (FSE)(**B**) of our patient. No abnormalities were noticed. (**C**,**D**): Fat-suppressed T2 sequences of cervical (**C**) and lumbar (**D**) MRI scans showing hypertrophy of hyperintense nerve roots (white arrows).

**Figure 2 brainsci-10-00282-f002:**
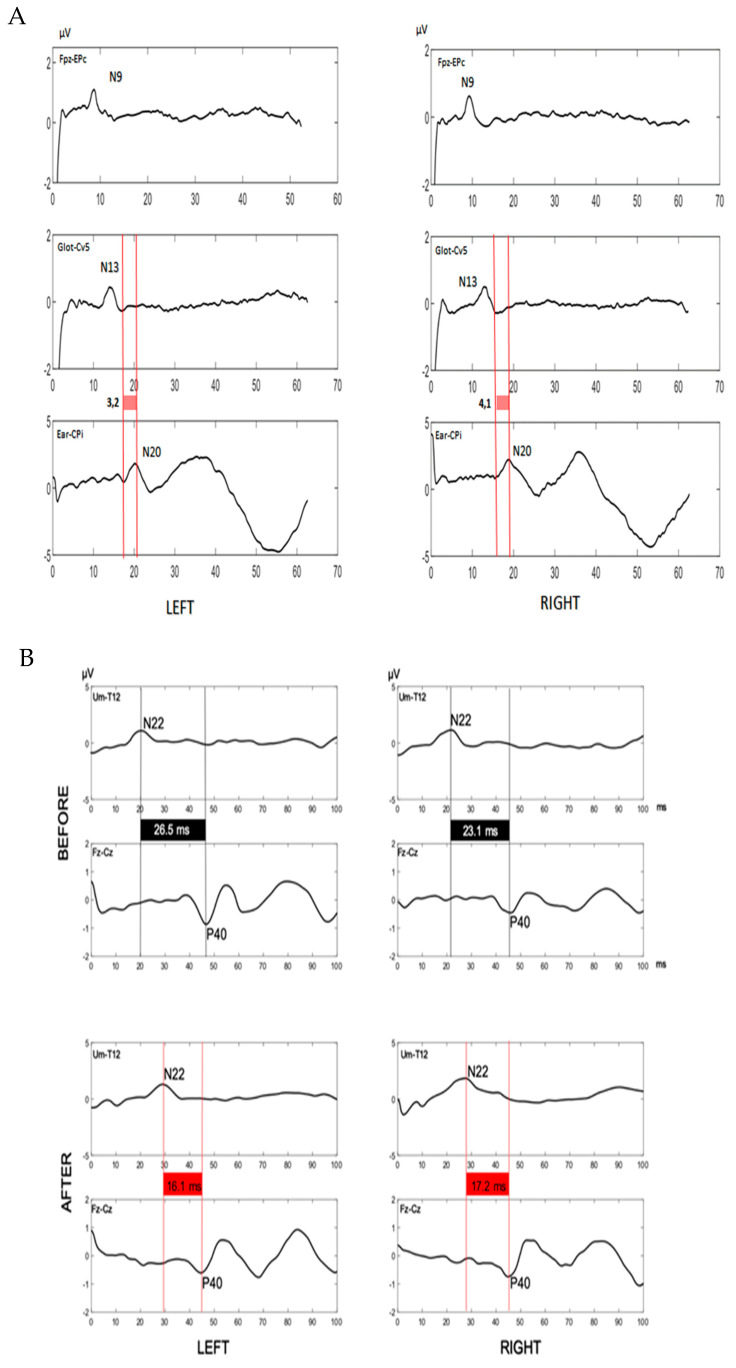
**Somatosensory evoked potentials**. (**A**). Somatosensory evoked potentials of upper limbs. Central conduction time upper limit in our laboratory: 6.2 ms. N9: brachial plexus somatosensory potential (upper limit in our laboratory: 11.6 ms). N13: spinal somatosensory potential (upper limit in our laboratory: 15.8 ms). N20: cortical somatosensory potential (upper limit in our laboratory: 21.6 ms). Fpz: forehead recording scalp electrode. Epc: Erb’s point dermal electrode. Glot: glottis dermal electrode. Cv5: spine of fifth cervical vertebra dermal electrode. Ear: auricular dermal electrode. CPi: centroparietal recording scalp electrode. All values were within the normal range before and after immunosuppressive therapy. (**B**). Somatosensory evoked potentials of lower limbs before and after corticosteroids. Somatosensory evoked potentials traces were displayed for left and right lower limbs. Please note the pathological values of the central conduction time (black boxes) before corticosteroids (upper traces) and their reduction and normalization (red boxes) after corticosteroids (lower traces). Furthermore, N22 latencies are normal before and pathological after corticosteroids. These data suggest that the dorsal column damage precedes the development of peripheral neuropathy and regress after immunomodulatory therapy. Central conduction time upper limit in our laboratory: 21.3 ms. N22 upper limit in our laboratory: 25.8 ms. N22: spinal somatosensory potential. P40: cortical somatosensory potential. Um: umbilicus dermal electrode. T12: spine of 12nd thoracic vertebra dermal electrode. Fz: fronto-median recording scalp electrode. Cz: centro-median recording scalp electrode.

**Figure 3 brainsci-10-00282-f003:**
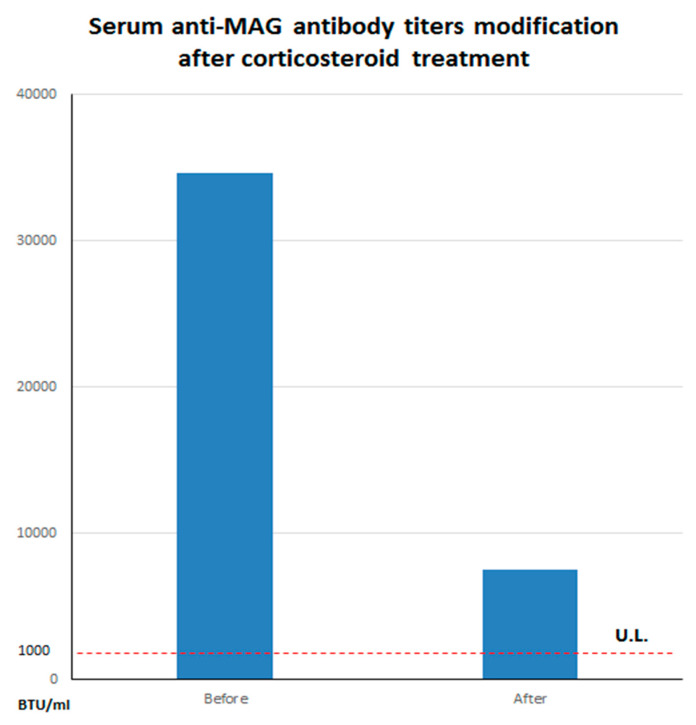
Serum anti-myelin associated glycoprotein (anti-MAG) titers modification after corticosteroid treatment. Please note the substantial decrease of serum antibody levels before (34,594 BTU/mL) and after (7520 BTU/mL) corticosteroid treatment. The dashed line refers to the upper limits (U.L.) of normal for our laboratory (~1000 BTU/mL).

**Table 1 brainsci-10-00282-t001:** Nerve conduction studies.

Nerve	First Evaluation	After Six Months	After Rituximab	Nerve	First Evaluation	After Six Months	After Rituximab
**Ulnar Right**		**cMAP**		**Ulnar Left**		**cMAP**	
DML (ms)	2.5	2.6	2.7	DML (ms)	2.1	2.5	2.2
MCV (m/s) below elbow to wrist	54	55	51	MCV (m/s) below elbow to wrist	55	59	53
Amplitude (mV)	8.5	8.2	8.3	Amplitude (mV)	8.2	8.3	8.1
F wave latency (ms)	26	24.3	24.7	F wave latency (ms)	25.1	25.4	25.2
**Ulnar Right**		**SNAP**		**Ulnar Left**		**SNAP**	
SCV (m/s) wrist to digit 5	52	51	50	SCV (m/s) wrist to digit 5	50	55	53
Amplitude (µV)	7.5	7.1	7	Amplitude (µV)	7.2	7.4	7.2
**Median Right**		**cMAP**		**Median Left**		**cMAP**	
DML (ms)	2.6	2.7	2.4	DML (ms)	2.5	2.8	2.6
MCV (m/s) elbow to wrist	58.3	59.1	58.7	MCV (m/s) elbow to wrist	59.4	58.6	57.6
Amplitude (mV)	6.5	6.2	5.9	Amplitude (mV)	6.8	6.4	5.8
F wave latency (ms)	24.2	24.5	24.3	F wave latency (ms)	24.7	24.6	24.8
**Median Right**		**SNAP**		**Median Left**		**SNAP**	
SCV (m/s) wrist to digit 3	35(*)	37(*)	37(*)	SCV (m/s) Wrist to Digit 3	38(*)	39(*)	40(*)
Amplitude (µV)	12.2	12.5	11.7	Amplitude (µV)	14.5	14.3	13.7
**Tibial Right**		**cMAP**		**Tibial Left**		**cMAP**	
DML (ms)	4.8	6.1(*)	5.0	DML (ms)	4.8	5.8 (*)	5.1
MCV (m/s) popliteal fossa to medial ankle	52	42	48	MCV (m/s) popliteal fossa to medial ankle	41	35.5 (*)	44.5
Amplitude (mV)	4.8	4.6	4.7	Amplitude (mV)	7.4	7.2	6.8
F wave latency (ms)	47.5	55.2(*)	51	F wave latency (ms)	48.1	60.2(*)	53.3
**Peroneal Right**		**cMAP**		**Peroneal Left**		**cMAP**	
DML (ms)	4.2	5.8(*)	4.1	DML (ms)	4.1	3.9	4.2
MCV (m/s) below fibula to ankle	45	47	46	MCV (m/s) below fibula to ankle	49	38(*)	48
Amplitude (mV)	2.4	2.3	2.1	Amplitude (mV)	3.2	3.2	3.1
**Sural Right**		**SNAP**		**Sural Left**		**SNAP**	
SCV (m/s) sural-posterior ankle	54	38.8(*)	50	SCV (m/s) sural-posterior ankle	53.7	44.6(*)	49
Amplitude (µV)	11.4	8.7(*)	8.7(*)	Amplitude (µV)	11.8	9.8(*)	9.8(*)

**Table 1: Nerve conduction studies of upper and lower limbs in our patient.** DML: Distal motor latency; MCV: Motor conduction velocity; SCV: Sensory conduction velocity; CMAP: Compound muscle action potential; SNAP: Sensory nerve action potential. Reference values for healthy subjects in our laboratory for motor conduction studies were the following: Ulnar motor nerve, DML 2.4 (±0.5) ms, MCV 57 (±8.4) m/s, CMAP 8.9 (±2.8) mV, record: Abductor digiti minimi; Median motor nerve, DML 2.9 (±0.16) ms, MCV 60.25 (±2.99) m/s, CMAP 8,5 (±2.8) mV, record: Abductor pollicis brevis; Peroneal motor nerve, DML 3.8 (±0.6) ms, MCV 48 (±7.6) m/s, CMAP 3.7 (±1.6) mV, record: Extensor digitorum brevis (EDB); Tibial motor nerve, DML 4.5 (±0.8) ms, MCV 48 (±7.6) m/s, CMAP 9.8 (±4.2) mV, record: Abductor hallucis brevis. Ulnar F wave latency: ≤32; Median F wave latency: ≤31; Tibial F wave latency: ≤56. Reference values for healthy subjects in our laboratory for sensory conduction studies were the following: Ulnar sensory nerve, SCV 55 (±5.1) m/s, SNAP 21.6 (±16.2) µV; Median sensory nerve, SCV 40 (±15) ms, Amplitude 21.8 (±10.7) µV; Sural nerve, SCV 53 (±5.4) m/s, SNAP 21.0 (±9.8) µV. (*) = Out of limit values.

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
