# Peer review of "Spinal Cord Impairment in Anti-Mag Neuropathy: Evidence from Somatosensory Evoked Potentials"

_brainsci, 2020, doi:10.3390/brainsci10050282_

Round 1

Reviewer 1 Report

This is an interesting case report of a patient suffering with anti-MAG neuroapthy and dysfunction of dorsal column conduction.  Although a case report, the study elegantly illustrates utility of SSEPs and establishes novel pathogenesis in what was regarded as a pure peripheral nerve disorder.  I have minor comments.

1.  Please include a summary of peripheral nerve conduction studies, perhaps in a table format. 

2. A figure of the MRI of brain and spinal cord, as well as nerve root hypertrophy, would be important.

3.  In the upper limbs, was the median or ulnar nerves studied?  please clarify. It would be good to perhaps include a figure of the upper limb SSEP.

Reviewer 2 Report

The present study report an anti-Mag positive patient presenting with clinical and neurophysiological evidence of spinal cord involvement, before developing peripheral neuropathy. Somatosensory evoked potentials demonstrated a prolonged central conduction time for lower limbs, suggesting a dorsal column involvement. the patient underwent corticosteroids treatment with a temporary remission of sensory symptoms and normalization of the conduction time.  The authors briefly mentioned that the patient developed multiradicoloneuropathy, which was successfully treated with rituximab. They finally discussed the potential role of anti-MAG antibodies in the pathophysiology of dorsal column damage. 

While there are some novel findings in this study that might make if of interest to readers of the journal, there are some concerning issues that should be resolved. Besides, the English language should be revised.

It would be helpful to present data in graphs, such as the serum titer of anti-MAG antibodies before and after corticosteroids treatment.

The authors discussed the outcomes reported by others after ritixumab treatment however for the case reported in this study they briefly mentioned that the patient presented clinical and ENG neuropathy improvement. 
